# Learning Realistic Sketching: A Dual-agent Reinforcement Learning Approach

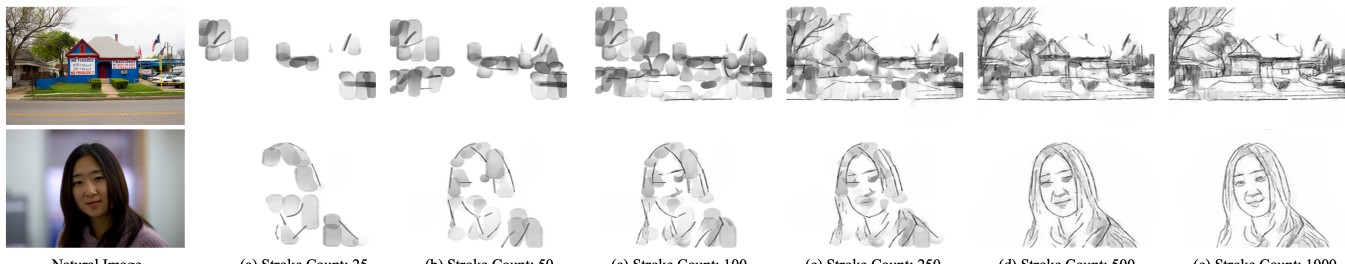

| Natural Image | (a) Stroke Count: 25 | (b) Stroke Count: 50 | (c) Stroke Count: 100 | (c) Stroke Count: 250 | (d) Stroke Count: 500 | (e) Stroke Count: 1000 |

**Figure 1: The image series displays the realistic sketching process of our work, transitioning from a natural image to a sketch.**

## ABSTRACT

This paper presents a pioneering method for teaching computer sketching that transforms input images into sequential, parameterized strokes. However, two challenges are raised for this sketching task: weak stimuli during stroke decomposition and maintaining semantic correctness, stylistic consistency, and detail integrity in the final drawings. To tackle the challenge of weak stimuli, our method incorporates an attention agent, which enhances the algorithm's sensitivity to subtle canvas changes by focusing on smaller, magnified areas. Moreover, in enhancing the perceived quality of drawing outcomes, we integrate a sketching style feature extractor to seamlessly capture semantic information and execute style adaptation at feature level, alongside a drawing agent that decomposes strokes under the guidance of the XDoG reward, thereby ensuring the integrity of sketch details. Based on dual intelligent agents, we have constructed an efficient sketching model. Experimental results attest to the superiority of our approach in both visual effects and perceptual metrics when compared to state-of-the-art techniques, confirming its efficacy in achieving realistic sketching.

## CCS CONCEPTS

• **Applied computing → Fine arts**.

## KEYWORDS

Digital Art, Realistic Sketching, Reinforcement Learning

*ACM MM, 2024, Melbourne, Australia*
© 2024 Copyright held by the owner/author(s). Publication rights licensed to ACM.
ACM ISBN 978-x-xxxx-xxxx-x/YY/MM
https://doi.org/10.1145/nnnnnnn.nnnnnnn

## 1 INTRODUCTION

Realistic sketching, a vital artistic medium, aims to capture the essence, appearance, and structure of a subject through rapid, informal, and simplified drawings. If computers are able to replicate this intricate process, it could demystify the art form, provide invaluable insights, and enhance the development of sophisticated drawing tools, thus supporting and enhancing human artistic endeavors.

We aim to explore teaching computers to sketch, with the goal of designing a machine drawing algorithm capable of simulating the human drawing process. As shown in Figure 1, the algorithm seeks to transform input images from pixel space into a set of sequential, parameterized strokes, capturing the essence, form, and structure of the subject.

However, teaching computers to sketch faces two major challenges. Firstly, how to address the issue of weak stimuli during stroke decomposition. Each stroke has minimal impact on the overall appearance of the canvas. The ability to accurately perceive subtle changes in the canvas before and after drawing will directly affect the algorithm's stroke decomposition and planning capabilities. Secondly, maintaining semantic correctness, stylistic consistency, and detail integrity simultaneously is challenging for drawing results. For sketching, these three requirements collectively determine people's perceptual experience.

To address the issue of weak stimuli in stroke sequential decomposition, current methods [13, 20, 37] typically adopt a strategy of grid-based partitioning, dividing the canvas uniformly into multiple regions, each utilizing the same number of strokes. During the actual drawing process, only one region is enlarged and drawn each time, and after the drawing is completed, it is then shrunk and pasted back onto the canvas. Since the area of each region is significantly smaller than that of the entire canvas and undergoes magnification, the visual differences before and after each stroke within the region become pronounced. This strategy significantly enhances the algorithm's ability to perceive the subtle changes to the canvas with each stroke.

Nevertheless, grid-based partitioning strategy also introduces two new problems: firstly, the uniform partition inevitably disrupts the continuity of some strokes; secondly, the uniform allocation

of strokes may lead to excessive redundancy in some regions and insufficient strokes in others. Consequently, these two problems severely constrain the visual quality of the resulting drawing.

For sketching, the semantic accuracy, stylistic consistency, and detail integrity of the drawing results determine people's perceptual experience. Some studies [22, 29] employ CLIP [24] or VGG perceptual losses [27] to measure the differences between input images and drawing results, or the consistency with sketch ground-truth, to address semantic issues. Stylistic consistency is maintained by manually setting stroke width and color, but the fixed width and color limit these methods to simple sketches, making them unsuitable for complex ones. Additionally, due to the lack of fine-grained loss, these methods often struggle to capture complete details. Other studies [13, 20, 37] address the issue of detail integrity by using distributional losses to measure the distribution differences between drawing results and ground-truth. However, distributional measurements often struggle to accurately distinguish the semantic importance of various details in images. Coupled with grid-based partitioning that disrupts overall semantics, this often results in the loss of important semantic details, consequently compromising both semantic accuracy and the integrity of details. For stylistic consistency, studies [17, 20, 37] use style transfer losses like the Gram matrix, but these are too simplistic to adequately ensure sketch-style drawing results.

To achieve more realistic and higher-quality drawing results, we need to address all three issues simultaneously. However, research in this area is still relatively scarce.

In this paper, to address the limitations of current methods, we propose a novel approach based on dual intelligent agents.

To address stroke interruptions and uniform stroke allocation issues caused by grid-based methods in handling weak stimuli, we introduce an attention agent. This agent utilizes input natural image data, current canvas conditions, and region selection history to determine the optimal position and size of the next drawing region, where more strokes are needed. In contrast to traditional grid-based techniques, our approach enables flexible stroke allocation by dynamically adjusting the drawing position and enhances the continuity of the stroke by adaptively setting the size of the drawing region. This innovative method not only addresses weak stimuli issues but also surpasses the limitations associated with conventional grid-based partitioning methods.

Furthermore, we address the complexities of achieving realistic sketching, encompassing semantic accuracy, stylistic consistency, and detail integrity, by decoupling semantic extraction and style transformation from the drawing process. We introduce a sketching style feature extractor designed to accurately capture semantic information from natural images at a feature level while executing style adaptation. Additionally, we propose a drawing agent focused on decomposing brush strokes during the drawing process. By incorporating distribution rewards and our newly introduced XDoG reward mechanism, this agent ensures the integrity of sketch details.

The outcomes of stroke decomposition experiments demonstrate the superiority of our method over current state-of-the-art (SOTA) techniques in terms of visual effects and perceptual metrics, when using the same number of strokes. This underscores the effectiveness of our approach in addressing the limitations of grid-based

methods in handling weak stimuli. Moreover, sketching experiments conducted on a diverse range of real-world scene images, spanning from portraits to architectural structures and landscapes, consistently showcase the superior performance of our method in both visual effects and perceptual metrics. Our method excels at capturing intricate semantic details and stylistic features, resulting in more realistic and lifelike drawing outcomes.

## 2 RELATED WORK

### 2.1 Style Transfer

As is well known, computers can transfer images from one style to another through image-to-image mapping. For instance, they can generate sketches from images through mathematical and geometric operations [6, 31]. In addition to that, the generation of sketches involves utilizing geometric features and occluding contours, along with integrating geometry-based methods with deep learning technologies [3, 8, 15, 19]. Learn2Generate [7] converts natural images into high-quality sketches using CycleGAN [36]. They added a geometric loss to predict depth information from sketches, and a semantic loss to ensure the generated image aligns with the original image's CLIP features [24].

However, these style transfer methods tackle the problem of image-to-image mapping, which significantly differs from the issue addressed in this paper. Our objective is to simulate the human drawing process by decomposing input natural images into a sequential and parameterized representation of sketch-style strokes. Existing style transfer algorithms struggle to achieve the goals of this paper through simple modifications.

### 2.2 Stroke Decomposition

Stroke decomposition aims to decompose a given target image into an ordered sequence of stroke parameters, achieving the transformation from the image to strokes. Initially, RNN-based algorithms explored the optimal stroke parameters, but limited to images with sparse strokes in small search spaces [11, 12, 34]. Recently, researchers have favored reinforcement learning due to its ability to maximize cumulative rewards throughout the drawing process[9, 13, 23, 25, 32, 35]. Modeling the drawing process as a Markov Decision Process, reinforcement learning effectively solves the decomposition problem in natural images. Although effective in determining stroke parameters, most methods work best with simple, uniform scenes rather than complex ones rich in details. This limitation arises from the problem of weak stimuli that obstructs further model convergence.

To mitigate this issue, methods like Learn2Paint [13], SNP [37], and PT [20] utilize a grid-based partitioning strategy to divide the canvas into smaller regions, improving the accuracy of stroke decomposition in complex scenes.

However, this strategy may diverge from the intuitive process of human drawing [28], potentially interrupting stroke continuity and semantic correctness when objects span multiple regions. This uniform stroke allocation approach often results in repetition in certain regions and insufficient detail in others. To address these challenges, this paper introduces an attention agent. This agent identifies key regions whose position and size adapt dynamically

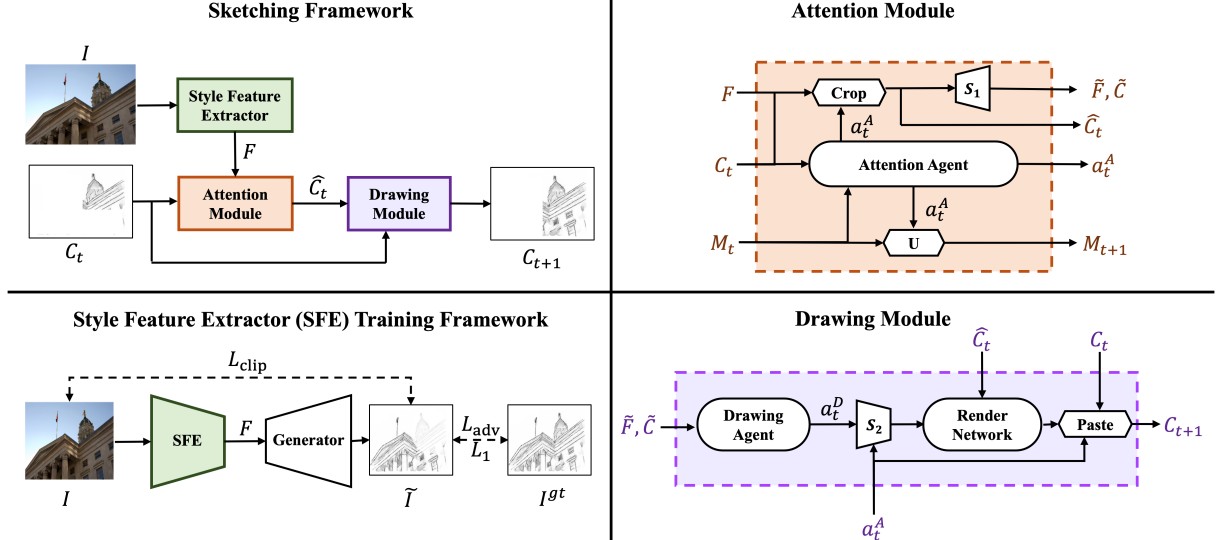

**Figure 2: Overview of the Sketching Framework: The top left illustrates the sequential interactions among the attention, extractor, and drawing components. The bottom left focuses on the style feature extractor (Section 3.2), the top right details the attention module (Section 3.3), and the bottom right describes the drawing module (Section 3.4).**

based on input natural images and the current canvas state, ensuring continuous strokes and their allocation as needed.

## 2.3 Style Transfer through Stroke Decomposition

Recent advancements like SNP [37] and RST [17] have merged stroke decomposition with the neural style transfer framework [10]. Furthermore, PT [20] has integrated AdaAttN [21] and LapStyle [18] into stroke inference to generate stylized drawings with varying colors and textures. However, sole reliance on basic style transfer losses such as the Gram matrix is insufficient for ensuring stylistic consistency between the drawing result and sketch ground truth, resulting in inaccurate strokes.

Researchers are now concentrating on converting natural images into sketches via image-to-stroke mapping. VSketch [22] and CLIPasso [29] each proposed an algorithm capable of translating natural images into strokes with an abstract sketch style. Yet, these methods with complex natural scenes, unable to capture complete details. The underlying problem is that loss functions based on CLIP [24] and VGG [27] allow the algorithm to capture only the broad semantics of the target object, lacking fine-grained loss for guidance. In contrast, our feature extractor can effectively capture style-adjusted features, facilitating the planning of sufficiently realistic sketching results.

## 3 METHOD

### 3.1 Overview

To address the challenges currently faced in sketching and overcome the limitations of existing approaches, we propose a novel sketching framework comprising three distinct components: a style feature extractor for extracting semantic and style-adjusted features, an attention module to determine the drawing regions, and a drawing module to infer stroke parameters for those regions.

The entire drawing process, as depicted in the top-left corner of Figure 2, begins with a natural image $I$ and an initial blank canvas $C_0$. Firstly, the style feature extractor extracts semantic and style-adjusted features $F$ from the natural image $I$. At each timestep $t$, the attention module selects an interest region $\hat{C}_t$, and then the drawing module applies appropriate strokes to this region, resulting in the next canvas $C_{t+1}$. After a series of numerous timesteps, the final sketch is obtained and denoted as $C_n$. The objective of the task is to minimize the difference between $C_n$ and $I$ in terms of semantics and details.

### 3.2 Style Feature Extractor

Our framework achieves precise semantic extraction and style adjustment through the utilization of a pre-trained style feature extractor. To train this extractor effectively, we integrate it with a generator to establish an image-to-image style transfer network, as shown in the bottom-left corner of Figure 2. The network is then trained using the following loss functions:

$$L_1 = \sum_i \|\tilde{I}_i - I_i^{gt}\|_1, \tag{1}$$

$$L_{clip} = \sum_i \|CLIP(I_i) - CLIP(\tilde{I}_i)\|_1, \tag{2}$$

$$L_{adv} = E_{I_i^{gt} \sim Q}[D(I_i^{gt})] + E_{\tilde{I}_i \sim P}[(1 - D(\tilde{I}_i))], \tag{3}$$

where $\tilde{I}_i$ and $I^{gt}$ are the generated sketch image and its sketch ground truth, $P$ and $Q$ symbolize the distributions of $\tilde{I}$ and $I^{gt}$, respectively. The $L_1$ loss is employed to gauge pixel-level disparities, promoting a comprehensive resemblance between $\tilde{I}_i$ and $I^{gt}$. Furthermore, the CLIP loss[24] enhances the semantic correctness

of $\tilde{I}_i$ by capturing semantic information from $I_i$. The term $L_{adv}$ represents the adversarial loss, which ensures stylistic consistency between $\tilde{I}_i$ and the $I^{gt}$, with $D$ acting as the discriminator to assess the degree to which the overall canvas resembles sketches.

By integrating the aforementioned losses, we establish the comprehensive loss function $L_G$ for training the style transfer network.

$$L_G = \lambda_1 L_1 + \lambda_{clip} L_{clip} + \lambda_{adv} L_{adv}, \tag{4}$$

where $\lambda_1 = 100$, $\lambda_{clip} = 10$, $\lambda_{adv} = 1$ are used in this paper.

This loss function $L_G$ effectively directs the feature extractor to precisely capture semantic information from natural images at a feature level while adapting styles.

### 3.3 Attention Module

To address the challenges faced by grid-based methods in handling weak stimuli, the attention module adopts an attention agent, illustrated in the upper right corner of Figure 2. This agent output action $a_t^A$ to dynamically adjust the position and size of region, thus preserving the advantages of grid-based methods while facilitating flexible stroke allocation and improving the continuity of single-stroke drawing.

Furthermore, to obtain the local canvas and feature maps required for stroke decomposition by the subsequent drawing agent, the attention module includes a Crop function, which is utilized to crop the drawing canvas and the feature map based on $a_t^A$. Subsequently, the cropped results undergo resizing to a predefined size using an operation $S_1$, resulting in the canvas $\tilde{F}$ and feature map $\tilde{C}$ needed by the drawing agent. Additionally, the Crop function also crops the canvas around the predicted drawing region center to obtain the cropped region $\hat{C}_t$, used for subsequent stroke rendering. The size of the cropped region $\hat{C}_t$ is set to 128 in this paper. And $M_t$ is the Historical Information Matrix used by the attention agent, and we will discuss its role and update method in the subsequent section dedicated to the attention agent.

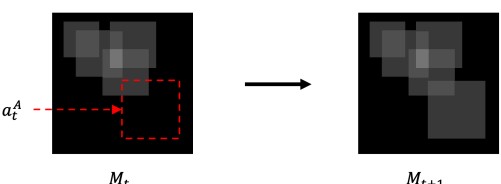

**Figure 3: We illustrate the History Information Matrix (HIM) evolution from $M_t$ to $M_{t+1}$.**

#### 3.3.1 Attention Agent.
Our attention agent relies on the current canvas and associated information to ascertain both the position and size of the next drawing region requiring more strokes. The action $a_t^A = \pi^A(s_t^A)$ and state $s_t^A$ of the attention agent at time $t$ are defined as follows:

$$a_t^A = (x, y, s), \tag{5}$$

$$s_t^A = (C_t, F, M_t, a_{t-1}^A, t), \tag{6}$$

where, $\pi^A$ is the learnt policy of this agent, $x, y$ denote the center coordinates of the selected region, $s$ represents the region's size. $C_t$

and $F$ correspond to the overall canvas and feature maps of the input natural image, respectively. We include $a_{t-1}^A$ from the previous step as it significantly influences current decision-making. To ensure the agent is sensitive to the full scope of historical information, we introduce a History Information Matrix (HIM) $M_t$, depicted in Figure 2. This matrix records the frequency with which each pixel on the canvas has been selected since the initial timestep. HIM $M_{t+1}$ is updated using a $U$ operation, where $M_{t+1}(x, y) = M_t(x, y) + 1$, if the position $(x, y)$ lies within the selected region; otherwise, $M_{t+1}(x, y) = M_t$, as illustrated in Figure 3.

Through the attention agent, our framework is capable of identifying regions that are either excessively painted or left uncovered, and it promotes the placement of strokes over more appropriate regions while ensuring a reasonable distribution of strokes across the canvas.

#### 3.3.2 Attention Reward.
The goal of the attention agent is to observe the current canvas and then determine the drawing region that should be focused on next. Given the absence of a direct supervisory signal to delineate the drawing region for the next timestep, we assess the precision of inferring this region by evaluating the disparity between the canvas $C_t$, and the ground truth, $I^{gt}$. If drawing on the target region brings the canvas closer to resembling the ground truth, a higher reward is awarded, thereby indicating a more accurate inference of the drawing region. Thus, the attention reward designed is based on the WGAN [2] loss reward $R^A$:

$$R^A = D_{wgan}(I^{gt}, C_t) - D_{wgan}(I^{gt}, C_{t-1}), \tag{7}$$

where $D_{wgan}(\cdot, \cdot)$ represents the discriminator, employed to assess the difference between the sketch ground truth and the overall canvas. $C_t$ and $C_{t-1}$ denote the canvas at the current and previous time steps, respectively.

### 3.4 Drawing Module

To swiftly and accurately infer strokes within drawing regions, our framework embeds an efficient drawing module, which senses the features and canvas within the region proposed by the attention agent to output the canvas rendered with new strokes. To enhance the detail drawing capability, this module introduces an XDoG drawing reward, precisely assessing the differences between the drawn image and the sketch ground truth.

As shown in the bottom-right corner of Figure 2, the drawing module includes an agent that receives features $\tilde{F}$ and the canvas $\tilde{C}$ within the drawing region determined by the attention agent to infer suitable stroke parameters $a_t^D$. These parameters are then scaled using the affine transformation function $S_2$, followed by rendering strokes onto $\hat{C}_t$. The render network establishes a differentiable mapping from stroke parameters $a_t^D$ and the current canvas $\hat{C}_t$ to the next canvas $\hat{C}_{t+1}$. This updated canvas $\hat{C}_{t+1}$ is then merged into the overall canvas $C_{t+1}$ using the Paste function. For details of the render network, please refer to the supplementary materials.

#### 3.4.1 Drawing Agent.
The drawing module contains a drawing agent, which has learnt the policy $\pi^D$ that maps the canvas $\tilde{C}$ and features $\tilde{F}$ to stroke parameters $a_t^D = \pi^D(s_t^D)$. The agent perceives

the drawing region, defined at time $t$ as follows:

$$a_t^D = (x_0, y_0, x_1, y_1, x_2, y_2, z_0, z_2, w_0, w_2, r, g, b), \quad (8)$$

$$s_t^D = (\tilde{C}, \tilde{F}, t), \quad (9)$$

where $a_t^S$ denotes the stroke parameters at time $t$, with $(x_0, y_0, x_1, y_1, x_2, y_2)$ as the control points of a quadratic Bézier curve. $(r, g, b)$ specifies the stroke's color, while $(z_0, z_2)$ and $(w_0, w_2)$ control the thickness and transparency at the start and end points, respectively.

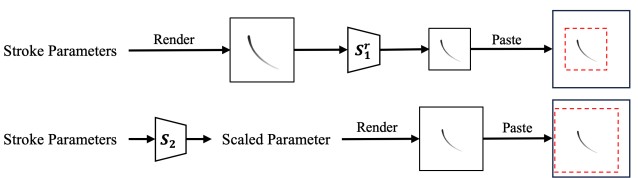

**Figure 4: The acceleration strategy that converts pixel-space scaling to stroke-space during the drawing process.**

*3.4.2 Rendering.* Upon completing the stroke decomposition, we need to render them onto the canvas. In the grid-based partitioning strategy, strokes are initially rendered on a local canvas, then the size of the local canvas is adjusted to fit the actual size by reverse operation $S_1^r$ of $S_1$, and finally, the adjusted canvas is pasted onto the main canvas, as illustrated in the upper image of Figure 4. The scaling operation during this process requires handling each pixel, with a time complexity of $O(n)$, where $n$ is the total number of pixels in the image. In practical applications, we found that this operation significantly increases training time.

Therefore, this paper proposes a new drawing process that shifts the scaling operation $S_2$ from pixel space to stroke parameter space, as depicted in the lower image of Figure 4. The time complexity of scaling operations in stroke space is $O(m)$, where $m$ is the total number of strokes. Since $m$ is much smaller than $n$, the new drawing process can significantly reduce computational complexity and greatly improve training efficiency.

The affine transformation $S_2$ is defined as follows:

$$S2 = \begin{bmatrix} 1 & 0 & x \\ 0 & 1 & y \\ 0 & 0 & 1 \end{bmatrix} \cdot \begin{bmatrix} f & 0 & 0 \\ 0 & f & 0 \\ 0 & 0 & 1 \end{bmatrix} \cdot \begin{bmatrix} 1 & 0 & -x \\ 0 & 1 & -y \\ 0 & 0 & 1 \end{bmatrix}. \quad (10)$$

The stroke parameter's coordinate system is modified by translating the origin from the bottom-left corner to the image's center $(x, y)$, followed by scaling the stroke size through multiplication by a factor $f$ based on $a_t^A$. Finally, the coordinate system of the scaled stroke is translated back, recentering it around the central point.

*3.4.3 Drawing Reward.* A specific reward mechanism is established for the drawing agent, aimed at guiding the agent to optimize strokes to closely resemble the ground truth. We also employ the WGAN discriminator score [2] as a reward function $R_{wgan}$. Unlike the attention reward $R^A$, $R_{wgan}$ focuses on the drawing region, calculating the distribution difference within the region between the drawn canvas $\hat{C}_t$ and the ground truth $\hat{I}^{gt}$.

However, a single $R_{wgan}$ alone cannot adequately guide the extraction of fine-grained information to ensure detail integrity. Therefore, we introduce a fine-grained reward function based on XDoG [31], denoted as $R_{xdog}$, to emphasize the details in sketches. XDoG [31] is an effective edge detection algorithm that exhibits remarkable results in capturing geometric information within images. By treating the edge map as a probability map $Z$ of size $L \times H \times W$ and using it to calculate the fine-grained reward, we enhance the agent's focus on edge information in the target image. The calculation of the fine-grained reward is as follows:

$$L_{xdog}(\hat{I}^{gt}, \hat{C}_t) = \frac{1}{L \times H \times W}\|(\hat{C}_t - \hat{I}^{gt}) \odot Z\|, \quad (11)$$

$$R_{xdog} = L_{xdog}(\hat{I}^{gt}, \hat{C}_{t-1}) - L_{xdog}(\hat{I}^{gt}, \hat{C}_t). \quad (12)$$

Ultimately, by integrating the two aforementioned reward functions, the total reward $R^D$ for the agent is obtained:

$$R^D = R_{wgan} + \lambda R_{xdog}, \quad (13)$$

where $\lambda$ represents the coefficient that balances the two rewards, set to 10 in practice.

## 3.5 Training Strategy

To further address the issue of weak stimuli, the agents employ a bundle action strategy, combining multiple consecutive operations into one action. The agent's bundle actions, represented by a vector $a = (a_1, a_2, \ldots, a_k)$, are observed and inferred every $k$ timesteps [4], thereby increasing the expected reward for each step and strengthening the reward signal. For both agents, empirical evidence has shown that setting the bundle size to 5 is most effective.

Due to the lack of direct supervision signals for inferring the drawing region and strokes, we optimize the models by employing a search-based approach through reinforcement learning. Multi-Agent TD3 (MATD3) [1] is utilized for training the two agents. In MATD3, distinct agents operate with unique policy gradients. In this paper, $\pi^A$ represents the policy of the attention agent, while $\pi^D$ signifies the policy of the drawing agent. $\theta^A$ and $\theta^D$ respectively denote their policy parameters, with the gradients as follows:

$$\nabla_{\theta^A} J(\theta^A) = E_{s^A}[\nabla_{\theta_A}\pi^A(a^A|s^A)\nabla_{a^A}Q^A(s^A, a^A, a^D)], \quad (14)$$

$$\nabla_{\theta^D} J(\theta^D) = E_{s^D}[\nabla_{\theta_D}\pi^S(a^D|s^D)\nabla_{a^D}Q^D(s^D, a^D)], \quad (15)$$

where $Q^A$ and $Q^D$ are the value functions of the attention and the drawing agents, respectively. In contrast to the standard MATD3, our implementation limits $Q^D$ to information within the drawing region, since global information does not improve stroke parameter precision and merely increases the number of training parameters.

## 4 EXPERIMENT

### 4.1 Dataset & Metric

For a comprehensive evaluation of the model's capability across various scenarios, we selected 2700 pairs of natural images and anime sketches from the MIT-Adobe FiveK dataset [5], with Learn2Generate creating the latter. Of these, 2100 pairs formed the training set, and 600 constituted the test set, ensuring a diverse and complete dataset. This selection offers a broad spectrum of visual features and structural elements, from basic shapes to complex textures, adequately

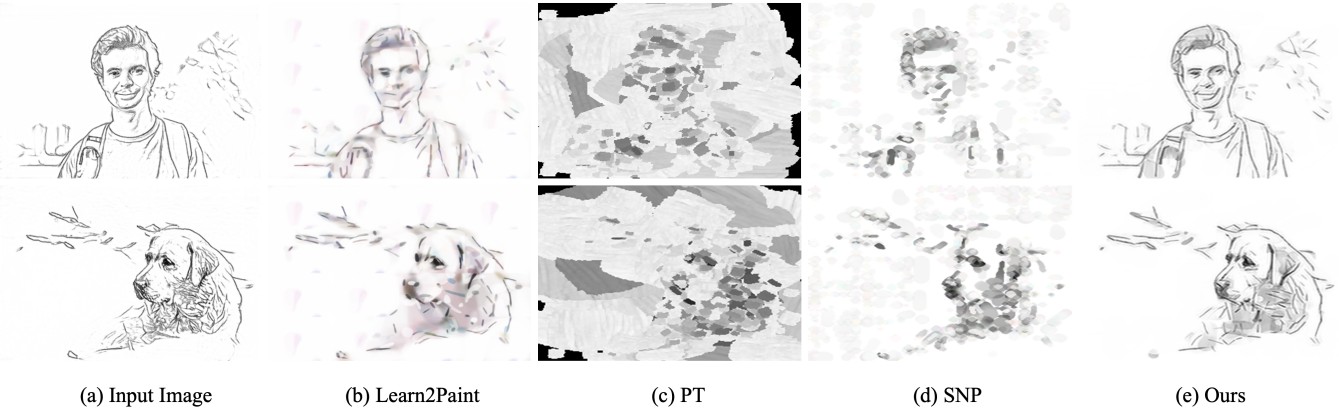

|  (a) Input Image | (b) Learn2Paint | (c) PT | (d) SNP | (e) Ours |

Figure 5: Comparison of stroke decomposition methods with 1000 strokes: (a) displays the input image, while (b) Learn2Paint [13], (c) PT [20], (d) SNP [37], and (e) our method achieves the closest resemblance to the input image.

Table 1: Comparison of existing drawing methods. (A) Can achieve style transfer; (B) Can produce sequential strokes; (C) Can ensure the semantic correctness; (D) Can ensure consistency with sketch style; (E) Can ensure the integrity of details.

| Methods | A | B | C | D | E |
|---|---|---|---|---|---|
| SNP [37] | ✔ | ✔ | ✘ | ✘ | ✘ |
| PT [20] | ✔ | ✔ | ✘ | ✘ | ✘ |
| Learn2Paint [13] | ✘ | ✔ | ✘ | ✘ | ✘ |
| CLIPasso [29] | ✔ | ✘ | ✔ | ✔ | ✘ |
| VSketch [22] | ✔ | ✔ | ✔ | ✔ | ✘ |
| RST [17] | ✔ | ✘ | ✘ | ✘ | ✘ |
| **Ours** | ✔ | ✔ | ✔ | ✔ | ✔ |

fulfilling the requirements for richness and variability in image textures and structures.

We employ PSNR [14] to measure the quality of image reconstruction, SSIM [30] for evaluating structural similarity, and LPIPS [33] to assess perceptual similarity, together providing a comprehensive evaluation of stroke decomposition. For realistic sketching, due to the stylistic gap between the input and output images, we augment our assessment by using the cosine similarity of CLIP [24] features to assess the semantic similarity with the natural image, and SIFID [26] to evaluate the style similarity with the sketch ground truth.

## 4.2 Baselines

We selected six SOTA methods as baselines for our experiments, and Table 1 summarizes their capabilities across six dimensions. For stroke decomposition, Learn2Paint [13], SNP [37], and PT [20] are chosen for their ability to sequentially output hundreds to thousands of strokes. In realistic sketching comparisons, we focus on methods with style transfer abilities like CLIPasso [29], SNP[37], VSketch[22], and RST [17]. PT [20] is excluded due to its unpublished code for the style transfer component.

Table 2: Comparative evaluation of stroke decomposition methods across PSNR, SSIM, and LPIPS metrics at varying stroke count.

| Metrics | Stroke Count | SNP | Learn2Paint | PT | Ours |
|---|---|---|---|---|---|
| PSNR ↑ | 200 | 17.38 | 18.41 | 11.23 | **18.64** |
|  | 500 | 17.86 | 19.15 | 11.57 | **19.58** |
|  | 1000 | 18.01 | 19.69 | 12.23 | **20.20** |
| SSIM ↑ | 200 | 0.5761 | 0.5768 | 0.3543 | **0.6102** |
|  | 500 | 0.5896 | 0.6269 | 0.3699 | **0.6768** |
|  | 1000 | 0.5749 | 0.6644 | 0.3686 | **0.7212** |
| LPIPS ↓ | 200 | 0.4633 | 0.4819 | 0.6374 | **0.3527** |
|  | 500 | 0.4340 | 0.3945 | 0.5918 | **0.2741** |
|  | 1000 | 0.4450 | 0.3410 | 0.5790 | **0.2419** |

## 4.3 Setup

For the computer sketching, a phased training strategy was employed. We first pre-trained the style feature extractor. Then, the drawing agent underwent training by randomly selecting positions. This allowed the agent to learn stroke parameters without attention guidance. Subsequently, the attention agent was trained while keeping the drawing agent's parameters fixed. Finally, both agents were jointly trained by MATD3. Training was conducted on a 256x256 resolution training set using the Adam optimizer [16] with a batch size of 32. All experiments were performed on the test set, rendering results with 1000 strokes without prior disclosure.

## 4.4 Stroke Decomposition Experiment

To evaluate the performance difference between the method proposed in this article and grid-based partitioning strategy based approaches in handling weak stimuli, we conducted stroke sequence decomposition experiments. Considering that some grid-based methods lack style transfer capabilities, we utilized sketches rather than natural images as inputs for the system.

As seen in Figure 5, PT [20] performs the poorest among the evaluated methods. This is attributed to its use of rigid, textured strokes without adjustable curvature parameters, limiting its effectiveness to oil painting scenarios and rendering it ineffective for sketch image stroke decomposition. While Learn2Paint [13]

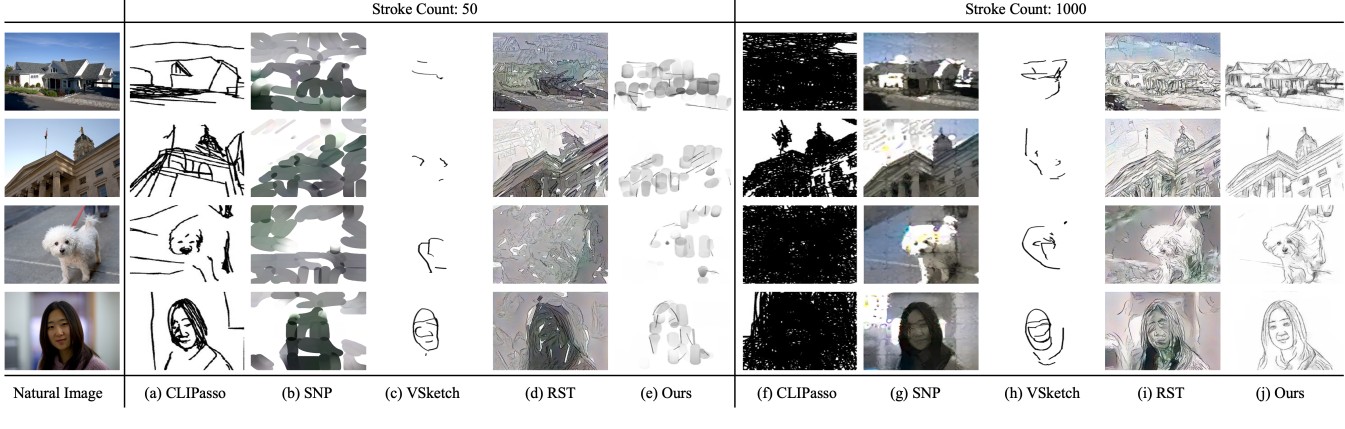

**Figure 6: Comparison of existing drawing methods with style transfer capabilities: Rows display outputs from natural images to sketches by (a) CLIPasso [29], (b) SNP [37], (c) VSketch [22], (d) RST [17], and (e) ours.**

**Table 3: Comparison with existing drawing methods with style transfer capabilities across different stroke counts.**

| Metrics | Stroke Count | SNP | RST | CLIPasso | VSketch | Ours |
|---|---|---|---|---|---|---|
| CLIP Cosine ↑ | 50 | 0.5889 | 0.5840 | **0.5869** | 0.5342 | 0.5806 |
| | 1000 | 0.6050 | 0.6196 | 0.5630 | 0.5361 | **0.6416** |
| SIFID ↓ | 50 | 40.657 | 31.687 | **25.937** | 57.939 | 34.066 |
| | 1000 | 49.076 | 21.823 | 95.002 | 41.529 | **5.777** |
| PSNR ↑ | 50 | 7.60 | 7.91 | 9.56 | 14.31 | **16.63** |
| | 1000 | 6.14 | 10.54 | 1.38 | 13.79 | **19.04** |
| SSIM ↑ | 50 | 0.3506 | 0.1951 | 0.4134 | 0.5240 | **0.5347** |
| | 1000 | 0.3281 | 0.2655 | 0.0471 | 0.5201 | **0.6737** |
| LPIPS ↓ | 50 | 0.7122 | 0.5949 | 0.5840 | 0.6176 | **0.5242** |
| | 1000 | 0.6954 | 0.4823 | 0.7394 | 0.5760 | **0.2523** |

and SNP [37] adeptly reconstruct input images, their results often suffer from artifacts due to stroke continuity inconsistencies and inadequate stroke allocation. Notably, SNP resorts to using coarse, wide strokes in regions where precise stroke parameter inference is challenging. In contrast, our method effectively captures fine details such as the dog's neck area with continuous strokes, devoid of interruptions evident in other methods.

Quantitatively, as shown in Table 2, our method outperforms others across various stroke counts, particularly excelling in all metrics. The table also indicates that methods employing grid-based partitioning exhibit sluggish performance improvement with increasing stroke counts, suggesting inadequate stroke allocation where needed and excessive allocation where unnecessary. In contrast, our method consistently demonstrates performance enhancement with increasing stroke counts, suggesting improved stroke allocation efficiency by the attention agent.

All the aforementioned results highlight the efficacy of the proposed attention agent in handling weak stimuli, as well as the drawing agent's proficiency in stroke decomposition and detailed drawing.

## 4.5 Realistic Sketching Experiment

We perform qualitative and quantitative analyses of our method compared to other baselines in realistic sketching. SNP [37] and

RST [17] treat natural images as content and sketch images as style inputs. Conversely, CLIPasso [29] directly optimizes from natural images, while VSketch [22] employs both natural images and sketches for training and inference. To provide a more comprehensive comparison, experiments have included scenarios with both 50 and 1000 strokes.

Comparing the performance of CLIPasso [29] and VSketch [22] under the settings of 50 and 1000 strokes, as depicted in Figure 6 and Table 3, it becomes apparent that they excel mainly in tasks requiring a small number of strokes, rather than in realistic sketches involving a multitude of strokes. VSketch[22] encounters challenges in completing drawing tasks satisfactorily, whether using 50 or 1000 strokes, due to limited stroke width and color, as well as a lack of semantic supervision and fine-grained detail guidance. CLIPasso[29] lacks the ability to decompose images into sequences of strokes, providing stroke parameters all at once. As stroke count increases, simultaneous optimization all strokes becomes exceedingly difficult, making CLIPasso incapable of completing drawing tasks with 1000 strokes.

From Figure 6, it's evident that whether drawing with 50 strokes or 100 strokes, both SNP [37] and RST [17] retain much of the content from natural images, indicating their struggles in adapting to sketch styles. This issue stems from their reliance on style loss derived from Gram matrices, primarily focused on color co-occurrence and frequency. Relying solely on this loss fails to ensure consistency between drawing results and sketch style, as indicated in Table 3.

As shown in Figure 6 and Table 3, our performance is inferior to that of the CLIPasso method [29] when using 50 strokes. This can be attributed to our method initially focusing solely on coarse contour drawing, resulting in a lack of finer details and consequently lower metrics. However, this aspect also reflects how our method mimics the human drawing process. We start by outlining the overall shape to establish the canvas layout, gradually refining it by adding details. Consequently, with 1000 strokes, our method achieved a state-of-the-art visual effect, far exceeding other baseline methods. As demonstrated in row 4, column j, our method accurately captures facial features, successfully achieving the goal of realistic sketching.

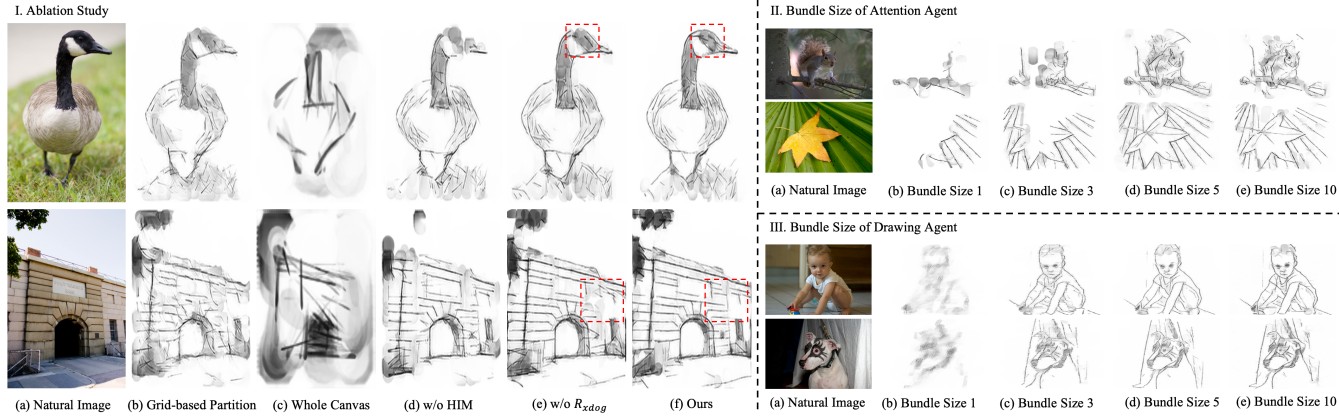

**Figure 7: An ablation study showcases the effects of different region partitioning strategies, the XDoG reward function, the History Information Matrix on sketch quality, and the impact of varying bundled action sizes.**

All the results mentioned above vividly demonstrate the effectiveness of our proposed method in simultaneously preserving semantic accuracy, stylistic coherence, and detail integrity in the drawing outcomes.

## 4.6 Ablation Studies

*4.6.1 Different Region Partitioning Strategies.* Grid-based partitioning strategy involves uniformly dividing the canvas into multiple grids and sequentially drawing within them. As illustrated in column b of Figure 7.I, this strategy allocates too many strokes to empty regions, resulting in blurred details and overlooking critical regions. The whole canvas approach, which draws directly on the entire canvas without focusing on any specific regions, as demonstrated in column c, overlooks all details due to the issue of weak incentives. In column f, our method's attention agent excels at identifying and detailing key regions, thereby ensuring the integrity of stroke shapes and the efficient allocation of stroke positions. Moreover, as shown in Table 4, it leads in the PSNR, SSIM, and LPIPS metrics, highlighting the crucial role of the attention module in realistic sketching.

*4.6.2 Without History Information Matrix M.* As demonstrated in column d of Figure 7.I, experiments showed that removing the History Information Matrix $M$ from the state space resulted in sketches with incomplete content coverage. After removing the History Information Matrix, all metrics declined as shown in Table 4, indicating that historical selection information effectively aids in determining the next region.

*4.6.3 Without $R_{xdog}$.* As shown in Column e of Figure 7.I, the removal of the XDoG reward results in the interruption or even absence of some strokes within the red dashed box. This further demonstrates that incorporating the XDoG reward significantly enhances stroke precision and boundary clarity, thereby ensuring the integrity of sketch details. Table 4 also reveals that the overall performance declines without the XDoG reward.

*4.6.4 Different Size of Bundle Actions.* We investigate the effect of different bundle sizes on sketch quality. As shown in Figures

**Table 4: Ablation studies examine sketching across varied settings, initially highlighting three region selection strategies: "GP" for grid-based partitioning, "WC" for whole canvas, and "AM" for attention module-determined regions. Further, "M" refers to the History Information Matrix in the attention agent, $R_{xdog}$ to XDoG reward, "BSA" to the attention agent's bundle size, and "BSD" to the drawing agent's bundle size.**

| GP | WC | AM | $M$ | $R_{xdog}$ | BSA | BSD | PSNR ↑ | SSIM ↑ | LPIPS ↓ |
|----|----|----|-----|-----------|-----|-----|--------|--------|---------|
| ✔ | | | - | ✔ | - | 5 | 17.73 | 0.573 | 0.330 |
| | ✔ | | - | ✔ | - | 5 | 15.85 | 0.496 | 0.622 |
| | | ✔ | | ✔ | 5 | 5 | 18.03 | 0.614 | 0.321 |
| | | ✔ | ✔ | | 5 | 5 | 16.62 | 0.499 | 0.435 |
| | | ✔ | ✔ | ✔ | 5 | 5 | 18.20 | 0.629 | 0.285 |
| | | ✔ | ✔ | ✔ | 1 | 5 | 16.14 | 0.553 | 0.512 |
| | | ✔ | ✔ | ✔ | 3 | 5 | 17.07 | 0.582 | 0.410 |
| | | ✔ | ✔ | ✔ | 10 | 5 | 17.99 | 0.609 | 0.324 |
| | | ✔ | ✔ | ✔ | 5 | 1 | 17.47 | 0.514 | 0.523 |
| | | ✔ | ✔ | ✔ | 5 | 3 | 18.43 | 0.615 | 0.319 |
| | | ✔ | ✔ | ✔ | 5 | 10 | 18.78 | 0.658 | 0.271 |
| | | ✔ | ✔ | ✔ | **5** | **5** | **19.04** | **0.674** | **0.252** |

7.II and 7.III, larger bundles enhance reward signals, thereby boosting training efficiency. However, excessively large bundles might complicate decision-making for individual steps and lead to an overabundance of strokes in single regions, disrupting the balance of stroke distribution.

## 5 CONCLUSION

This paper introduces a framework with two intelligent agents: an attention agent for adaptive stroke allocation and a drawing agent for stroke parameter inference. Automatic determination of drawing region positions ensures the allocation of strokes as required, while dynamic adjustment of region size maintains stroke continuity. The style feature extractor to capture semantic and style-adjusted features, and the drawing agent achieves more detailed drawing capabilities through the XDoG drawing reward. Our method exceeds current state-of-the-art techniques in stroke decomposition and realistic sketching across various scenes, showcasing advanced capabilities in computer sketching.

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
