# OpenReview forum: "Learning Realistic Sketching: A Dual-agent Reinforcement Learning Approach"
_acmmm.org/ACMMM/2024/Conference — MM2024 Poster_

### Official Review · Reviewer_1puY · 2024-05-23

**Rating:** 5
**Confidence:** 1

**Summary:**

This paper proposes a novel approach to the image-to-sketch task. The authors identify two main problems: resolving weak stimuli in stroke decomposition and improving the image quality. To address the former, they propose leveraging an attention agent focusing on smaller magnified areas. Regarding the latter, they introduce a sketching style feature extraction to capture semantic information and a drawing agent to decompose strokes under the guidance of the XDoG reward.

**Strengths:**

- easy to follow
- quantitative and qualitative metrics confirm the claims of the paper
- extensive supplementary

**Limitations:**

- more qualitative examples in the appendix could help the readers to evaluate the model better

**Suitability:**

3

---

### Official Review · Reviewer_XarH · 2024-05-25

**Rating:** 3
**Confidence:** 2

**Summary:**

The authors propose a method to transform input images into sequential, parameterized strokes. They introduce an attention agent to address the challenge of weak stimuli and a sketching style feature extractor for better semantic information and style adaptation. The results show improved visual and perceptual effects.

**Strengths:**

1. The authors claim this work is pioneering in teaching computers to sketch by transforming input images into sequential, parameterized strokes. Does this pioneering aspect refer to being the first work on this specific task or another novel contribution?
2. The method demonstrates superior performance in experiments, both in terms of visual effects and perceptual metrics.

**Limitations:**

1. The authors use five metrics—SSIM, PSNR, CLIP, SIFID, and LPIPS—to evaluate the method. However, these metrics primarily measure similarity on the pixel or perception level. The claimed challenges involve disruptions in the continuity of strokes and excessive redundancy in some regions with insufficient strokes in others. These metrics do not effectively evaluate the performance concerning these specific challenges.
2. There are awkward paragraph breaks in sections #141, #144, and #146.
3. Based on the above concerns, my initial rating is borderline reject. However, I am open to increasing my rating if the authors can address these issues.

**Suitability:**

3

---

### Official Review · Reviewer_p9Xo · 2024-05-25

**Rating:** 4
**Confidence:** 1

**Summary:**

This paper proposes dual intelligent agents to address two major challenges in teaching machines to sketch using stroke decomposition: weak stimuli during the stroke decompositions and maintaining semantic correctness, stylistic consistency, and detail integrity in the final drawings. An attention agent, the first intelligent agent, determines the optimal positions and size of the drawing region on the canvas, which differs from existing uniform grid-based partitioning strategies where stroke continuity and semantic correctness are often ignored. The drawing agent predicts parameterized strokes and renders them only to the regions of interest determined by the attention agent. The authors provide a variety of experimental results including ablation studies, showcasing the superiority of the proposed method over selected baseline models.

**Strengths:**

1. The problems to be addressed in the paper are clearly defined and described, and each module of the proposed method is well-designed to suit the problem.

2. There are many details in terms of model architecture, training, and rendering (e.g., the style feature extractor, the XDoG reward function, and the stroke-space scaling), which I believe other existing works could benefit from it as well.

3. The proposed method has been properly tested on two different tasks, and ablation studies demonstrate the effectiveness of each module.

**Limitations:**

1.The proposed method performs worse on some metrics than CLIPasso in realistic sketching experiments (Tab. 3). The authors claimed that this is because the proposed method initially focuses solely on coarse contour drawing. However, it is hard to tell if the resulting images contain the contour of the entire objects (Fig. 6), rather the contour tends to cover the semantically meaningful parts of the objects (e.g., a dog’s nose) first. That is, the proposed method appears to require sufficient timesteps to draw an image including complete objects. This can be seen more clearly in Fig. 6 in the supplementary materials.

2. In a similar vein, it would be helpful for users if the authors could provide guidance on the number of steps to draw images including complete objects.

3. There may be concerns about the rendering speed. How much time does it take to render a single image with 500 or 1,000 strokes? It would be better to compare the rendering time of the proposed methods against the baseline models.

4. From my understanding, this work primarily focuses on the sketching task. Can it also be applied to the painting task? If so, how good is the proposed method compared to the baseline models on the painting task?

**Suitability:**

2

---

### Official Review · Reviewer_NDSc · 2024-05-25

**Rating:** 3
**Confidence:** 3

**Summary:**

The paper presents a novel method for teaching computer sketching that converts input images into parameterized strokes. It addresses two key challenges: weak stimuli during stroke decomposition and maintaining semantic correctness, stylistic consistency, and detail integrity. To enhance sensitivity to canvas changes, an attention agent is used, focusing on smaller areas. Additionally, a sketching style feature extractor captures semantic information and performs style adaptation, while a drawing agent ensures stroke detail integrity under the guidance of the XDoG reward.

**Strengths:**

- **Dual-agent approach:** the paper introduces a novel method using dual intelligent agents, an attention agent for adaptive stroke allocation and a drawing agent for stroke parameter inference, enhancing stroke continuity and allocation efficiency.
- **Visual and perceptual metrics:** extensive experiments demonstrate the method's superiority over state-of-the-art techniques in terms of visual effects and perceptual metrics.

**Limitations:**

- **Clarity of notation:** the order in which names and components is off, sometimes acronyms compare before they are defined, e.g. XDoG in the abstract or $\alpha_t^A$ (L366).
- **Claim support:** the first part of the introduction is not properly supported by citations, e.g. "[...] it could demystify the art form, provide invaluable insights [...]" (L82).
- **Missing references and comparisons:** for example, the grid-based partition problem (L113-119) has been tackled before by the community, in particular by [1], a work not discussed by the authors. Another missing reference and comparison is Clipascene [2], an improvement of the referenced CLIPasso. For recent applications of Neural Painting in creative content creation see also [3].

[1] Hu, Teng, et al. "Stroke-based Neural Painting and Stylization with Dynamically Predicted Painting Region." Proceedings of the 31st ACM International Conference on Multimedia. 2023.
[2] Vinker, Yael, et al. "Clipascene: Scene sketching with different types and levels of abstraction." Proceedings of the IEEE/CVF International Conference on Computer Vision. 2023.
[3] Peruzzo, Elia, et al. "Interactive Neural Painting." Computer Vision and Image Understanding 235 (2023): 103778.

**Suitability:**

2

---

### Meta-Review · Area_Chair_TTLW · 2024-07-02

**Recommendation:** Accept (Poster)
**Confidence:** 5

**Metareview:**

The paper introduces a novel method for teaching computers to sketch, specifically transforming input images into sequential, parameterized strokes. All reviewers have reached a consensus on accepting this paper. Therefore, I recommend its acceptance. Please ensure that additional experiments mentioned by reviewers during the rebuttal are incorporated into the final version to further strengthen the paper.